# Novel Splicing Mutation in *MTM1* Leading to Two Abnormal Transcripts Causes Severe Myotubular Myopathy

**DOI:** 10.3390/ijms231810274

**Published:** 2022-09-07

**Authors:** Luca Bosco, Daniela Leone, Laura Costa Comellas, Mauro Monforte, Marika Pane, Eugenio Mercuri, Enrico Bertini, Adele D’Amico, Fabiana Fattori

**Affiliations:** 1Genetics and Rare Diseases Research Division, Unit of Neuromuscular and Neurodegenerative Disorders, Department of Neurosciences, Bambino Gesù Children’s Hospital, IRCCS, Viale S. Paolo 15, 00146 Rome, Italy; 2Department of Science, University “Roma Tre”, Viale Marconi 446, 00146 Rome, Italy; 3Centro Clinico Nemo, Fondazione Policlinico Universitario A. Gemelli IRCCS-Università Cattolica del Sacro Cuore, Largo A. Gemelli 8, 00146 Roma, Italy; 4Pediatric Neurology, Vall d’Hebron Institut de Recerca (VHIR), Hospital Universitari Vall d’Hebron, Vall d’Hebron Barcelona Hospital Campus, Passeig de la Vall d’Hebron, 119-129, 08035 Barcelona, Spain; 5UOC di Neurologia, Fondazione Policlinico Universitario A. Gemelli IRCCS, Largo A. Gemelli 8, 00146 Rome, Italy; 6Pediatric Neurology, Department of Woman and Child Health and Public Health, Fondazione Policlinico Universitario A. Gemelli IRCCS-Università Cattolica del Sacro Cuore, Largo A. Gemelli 8, 00146 Rome, Italy

**Keywords:** *MTM1*, splicing, XLMTM, myotubular myopathy, novel mutation, NGS

## Abstract

X-linked myotubular myopathy (XLMTM) is a severe form of centronuclear myopathy, characterized by generalized weakness and respiratory insufficiency, associated with pathogenic variants in the *MTM1* gene. NGS targeted sequencing on the DNA of a three-month-old child affected by XLMTM identified the novel hemizygous *MTM1* c.1261-5T>G intronic variant, which interferes with the normal splicing process, generating two different abnormal transcripts simultaneously expressed in the patient’s muscular cells. The first aberrant transcript, induced by the activation of a cryptic splice site in intron 11, includes four intronic nucleotides upstream of exon 12, resulting in a shift in the transcript reading frame and introducing a new premature stop codon in the catalytic domain of the protein (p.Arg421SerfsTer7). The second aberrant *MTM1* transcript, due to the lack of recognition of the 3′ acceptor splice site of intron 11 from the spliceosome complex, leads to the complete skipping of exon 12. We expanded the genotypic spectrum of XLMTM underlying the importance of intron–exons boundaries sequencing in male patients affected by XLMTM.

## 1. Introduction

X-linked myotubular myopathy (XLMTM, OMIM 310400) is the most common and severe form of centronuclear myopathy [1], affecting approximately 1/50,000 male newborns [2]. The affected babies manifest severe hypotonia and respiratory insufficiency at birth [3], and the majority of them die during the first months of life [2]. Muscle biopsy generally shows central nuclei that are larger than normal and have a vesicular appearance [4], and it is often characterized by the presence of “necklace fibers” that are peculiar myofibrillar aggregates surrounded by mitochondria, sarcoplasmic reticulum and glycogen granules [5].

XLMTM is caused by mutations in the *MTM1* gene, located in the distal long arm of the X chromosome (locus Xq28), coding for myotubularin, one of the catalytically active phosphatases of the myotubularins superfamily, which plays different roles in muscular cells. It heterodimerizes with MTMR14, leading to skeletal muscle maintenance and Ca^2+^ homeostasis, and it binds directly desmin, regulating intermediate filament assembly and architecture [6]. Structurally, myotubularins are constituted by four characteristic main domains: the catalytic protein tyrosine phosphatase (PTP), the predicted glucosyltransferases, Rab-like GTPases activators and myotubularins (GRAM), the Rac-induced recruitment domain (RID) and the SET-interaction domain (SID) [7].

Here, we described the case of a 3-month-old child affected by a severe form of XLMTM and a poor prognosis, harboring a novel hemizygous *MTM1* intronic variant. We demonstrated that this variant causes an aberrant splicing mechanism, generating two different abnormal transcripts simultaneously expressed in muscular cells.

## 2. Results

### 2.1. Clinical Findings

Proband is the first male who was born to non-consanguineous parents. The pregnancy was complicated by polyhydramnios, although the preserved fetal movements were not perceived to be decreased. He was born from induced uneventful delivery at 39 + 6 weeks of gestation. At birth, the baby presented with a respiratory distress with pulmonary hypertension requiring invasive mechanical ventilation. At the age of one month, he was extubated and supported by non-invasive ventilation only during sleeping hours for 3 months. Following an acute respiratory infection, he needed to be intubated again, and, after multiple failed extubation attempts, a tracheostomy was placed at age 5 months. He was then fed through a nasogastric tube up to the age of 3 months, when a gastrostomy was placed. Since birth, he showed a generalized hypotonia, suggesting a congenital myopathy. Muscular biopsy of the left quadriceps showed features of centronuclear myopathy (Figure 1). The abdominal echography showed a renal lithiasis and right cryptorchidism. During the first months of life, a limited downward gaze, macrocephaly and dolichocephaly were noticed. At the age of 8 months, he was able to sit with support, with a partial head control, and had anti-gravity movements at four limbs. The patient died at 13 months old due to respiratory failure.

### 2.2. Molecular Data

Targeted resequencing using NGS custom panel allowed us to identify in our patient the novel hemizygous c.1261-5T>G variant affecting the acceptor splice site of exon 12 in the *MTM1* gene (NM_000252.3) as the only significant variant related to the patient’s phenotype. Sanger sequencing confirmed NGS data in Proband, and a segregation analysis showed that the healthy patient’s mother carried the variant in heterozygosity (Figure 2). The nucleotide change c.1261-5T>G is not reported in any publicly available human variation database (i.e., dbSNP146, 1000 Genomes, Exome Aggregation Consortium (ExAC), NHLBI Exome Sequencing Project (ESP) Exome Variant Server and gnomAD), and in silico bioinformatics pipelines predict this variant as negatively influencing the splicing mechanisms (SPiCE Probability: High (1)/VarSEAK class 5).

The transcript analysis from total RNA extracted from muscular biopsy revealed two different abnormal transcripts simultaneously expressed in the patient’s muscular cells, confirming that the new c.1261-5T>G variant causes aberrant splicing processes. Indeed, sequencing a fragment of *MTM1* cDNA encompassing exon 8 to 12 (Amplicon A1_Figure 3A) revealed the inclusion of four intronic nucleotides (UCAG) included upstream of exon 12 (Figure 3B), causing a shift in the transcript reading frame and resulting in a premature stop codon introduction in the catalytic PTP domain of myotubularin (p.Arg421SerfsTer7). This finding confirms that the c.1261-5T>G variant causes the activation of a cryptic acceptor splice site in intron 11, which is recognized by the spliceosome complex.

On the other hand, sequencing the *MTM1* cDNA fragment encompassing exon 11 to 14 (Amplicon A2_Figure 3A), allowed us to discover that the novel *MTM1* splicing variant likely prevents the spliceosome complex from recognizing the 3′ acceptor splice site of intron 11, causing the skipping of exon 12 from the mature transcript (Figure 3C) and resulting in the in-frame deletion of the region between Arg421 and Gln451 residues.

## 3. Discussion

Here, we report on a novel hemizygous splicing variant in *MTM1* resulting in a severe form of XLMTM with fatal prognosis. The c.1261-5T>G variant, identified in a 3-month-old child with a clinical and histological diagnosis of centronuclear myopathy, affects a conserved nucleotide in the consensus 3′ splice site of intron 11. The sequencing of *MTM1* cDNA obtained from RNA extracted from muscle biopsy in our patient revealed that the novel c.1261-5T>G variant interferes with the normal splicing process of mature *MTM1* mRNA, generating two different abnormal transcripts simultaneously expressed in the patient’s muscular cells. The first aberrant transcript, induced by the activation of a cryptic splice site in intron 11, includes four intronic nucleotides upstream of exon 12, resulting in a shift in the transcript reading frame that causes the introduction of a new premature stop codon seven triplets downstream of the Arginine 421 residue. The second aberrant *MTM1* transcript, due to the lack of recognition of the 3′ acceptor splice site of intron 11 from the spliceosome complex, leads to the complete skipping of exon 12, causing, if the protein is translated, the in-frame deletion of the region between Arg421 and Gln451 residues within the PTP domain of the protein.

More than 400 *MTM1* mutations in XLMTM have been described so far [6] widespread over all protein domains, meaning that there are no mutational hot spots in this gene [8]. According to the *MTM1*-LOVD database, the most common *MTM1* variants affecting the PTP domain are the p.Arg421Ter and p.Arg421Asn in exon 12, located in the same Arg421 residue, and the splicing c.1261-10A>G variant, affecting the 3′ acceptor splice site of intron 11 and causing the insertion of three aminoacids (p.Ser420_Arg421insPheIleGln) in the core of the PTP active site of the protein. These mutations, as the c.1261-5T>G variant in our patient, are associated with a severe disease outcome in affected males [9]. Functional data regarding the pathogenic mechanisms underlying these variants are not reported. However, it is known that, generally, nonsense mutations or deletions in the PTP domain result in the loss of myotubularin expression. Otherwise, missense mutations in the phosphatase domain are expected to inactivate the putative enzymatic activity of myotubularin [10,11], resulting in a clinical picture similar to that seen with nonsense variants, or may instead act by altering protein folding and stability [12]. Thus, it is presumed that the splice c.1261-5T>G variant, generating two aberrant transcripts affecting the catalytic PTP domain, likely impairs the synthesis of functional myotubularin and contributes to this severe form of XLMTM, leading to the loss of phosphatase activity (mainly due to the more represented p.Arg421_Gln451del transcript) or to the production of truncated protein (mainly due to the less abundant p.Arg421SerfsTer7 transcript), which is probably easily degraded. This could lead to abnormal dephosphorylation of phosphatidylinositol 3-phosphate and phosphatidylinositol 3,5-bisphosphate and subsequent generation of abnormal trafficking of the effector proteins of the endosomal–lysosomal pathway [13], generating a misplacement of organelles and fiber type alterations [14].

To date, more than 50 *MTM1* splicing variants, accounting for about 20% of XLMTM cases, have been classified as pathogenic or likely pathogenic in the *MTM1*-LOVD database, and RT-PCR analysis from muscle cDNA has been used to confirm the effect of splice mutations in most cases. Except for the recurrent c.1261-10A>G variant in intron 11 and for two deep intronic *MTM1* variants in introns 7 and 13, respectively [6,15,16], most of the *MTM1* splicing variants involve canonical GT and AG dinucleotides for donor and acceptor sites and have been found in isolated families (LOVD database). Only the splice c.529-1G>T variant in intron 7, identified in a patient with mildly progressive myotubular myopathy despite a strong reduction in the protein level, determined complex splicing rearrangements, leading to multiple abnormal splicing transcripts in the *MTM1* mRNA [17]. The c.1261-5T>G variant in our patient represents the second *MTM1* mutation leading to different abnormal *MTM1* transcripts and expands the genotypic spectrum of XLMTM, pointing out the importance of intron–exons boundaries sequencing and deepening studies on the patient’s cDNA for detecting any intronic variants leading to aberrant splicing mechanisms. These approaches will increase the rate of further genetic diagnoses of XLMTM patients, considering that the majority of intronic mutations in the *MTM1* gene are frequently related to a more severe form of XLMTM [8].

## 4. Materials and Methods

### 4.1. Muscle Biopsy

Under general anesthesia, a muscle sample was taken from the quadriceps by surgical excision. The tissue was snap frozen in liquid-nitrogen-cooled isopentane. Then, ten micrometers thick cryostat sections were stained using standard histochemical protocols [18].

### 4.2. Genetics Analysis

Genomic DNA was extracted from peripheral blood samples containing EDTA through the Qiagen protocol after receiving informed consent. Next-generation sequencing analysis was performed using the target enrichment method on Illumina platform with a uniquely customized panel related to muscular diseases, including 148 genes, as already described [19]. The patient’s DNA sample was enriched using custom probes with the Nextera Rapid Capture Custom Enrichment Kit (Illumina, San Diego, CA, USA) following the manufacturer’s instructions. DNA capture, enrichment and paired-end sequencing with the read length of 151 bp were performed using Illumina MiSeq with a sequencing depth of 100 X. The Illumina VariantStudio v3.0 data analysis software (Illumina, San Diego, CA, USA) was used to annotate the variants. Conventional Sanger sequencing was performed using ABI 3130xl capillary sequencer (Applied Biosystem, Thermo Fisher Scientific, Carlsbad, CA, USA) to confirm the novel variant identified by NGS in Proband and in his relatives.

### 4.3. Transcript Analysis

Total RNA was extracted from the patient’s muscular biopsy using Total RNA Purification Plus Kit (Norgen Biotek, Thorold, ON, Canada). Reverse transcription PCR (LUNA RT-PCR protocol-EUROCLONE) was performed using 500 ng of RNA template, and the cDNA obtained was amplified through primers specifically designed (Table 1) for generating different amplicons. The sizes of PCR products were confirmed by agarose 1.5% gel electrophoretic run, and conventional Sanger sequencing was performed.

### 4.4. In Silico Analysis

In silico supports for prediction of RNA splicing come from different bioinformatics tools, which are available online: Splicing Prediction Pipeline (SPiP; https://sourceforge.net/projects/splicing-prediction-pipeline/, 25 February 2022) and VarSEAK (https://varseak.bio/, varSEAK © 2022, 4 September 2022).

## Figures and Tables

**Figure 1 ijms-23-10274-f001:**
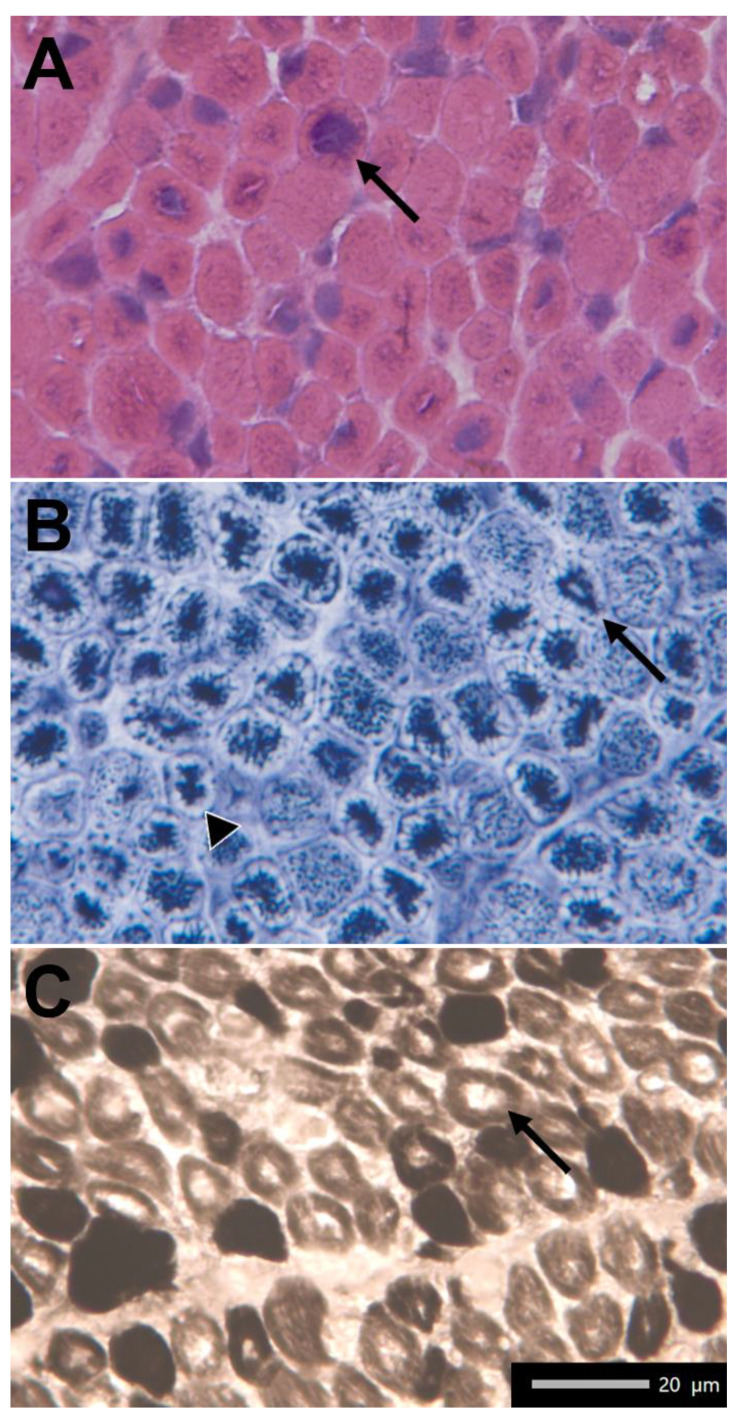
(**A**) Hematoxylin–Eosin. Slight increase in perimysial connective tissue. Enlarged, single central nuclei in many muscle fibers (arrow). (**B**) NADH-TR. Dark staining in the central region of many fibers with clear rim around edges (arrowhead) and perinuclear rings of mitochondria accumulation (arrow). (**C**) ATPase stain, pH 9.4. Slight predominance of type I fibers, smaller than type II, with presence of clear region in the center of many fibers (arrow).

**Figure 2 ijms-23-10274-f002:**
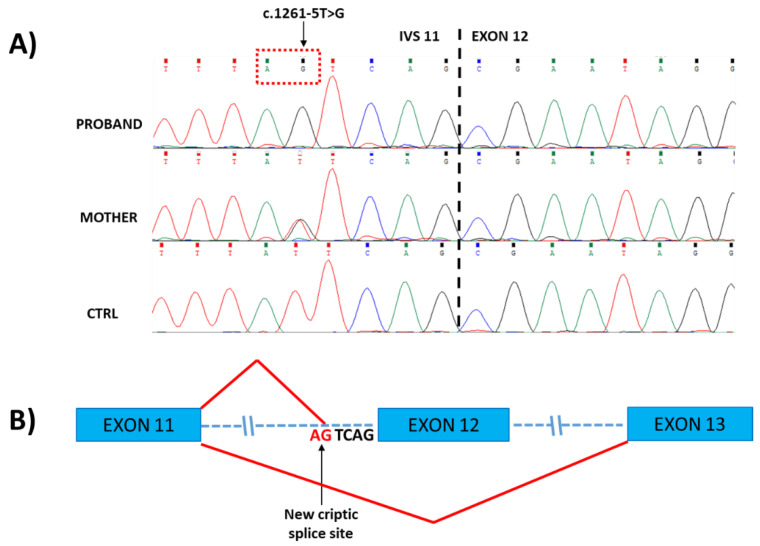
(**A**) Sanger sequencing of genomic DNA from Proband and his mother, showing the localization of the c.1261-5T>G variant in IVS11 inherited from the heterozygous mother. (**B**) Schematic representation of patient’s abnormal splicing processes, generating two different transcripts simultaneously expressed in muscular cells: the first aberrant transcript is generated by the activation of a cryptic splice site in intron 11 (indicated by the arrow), leading to the insertion of 4 bp (UCAG) upstream of exon 12; in the second aberrant transcript, exon 12 is skipped.

**Figure 3 ijms-23-10274-f003:**
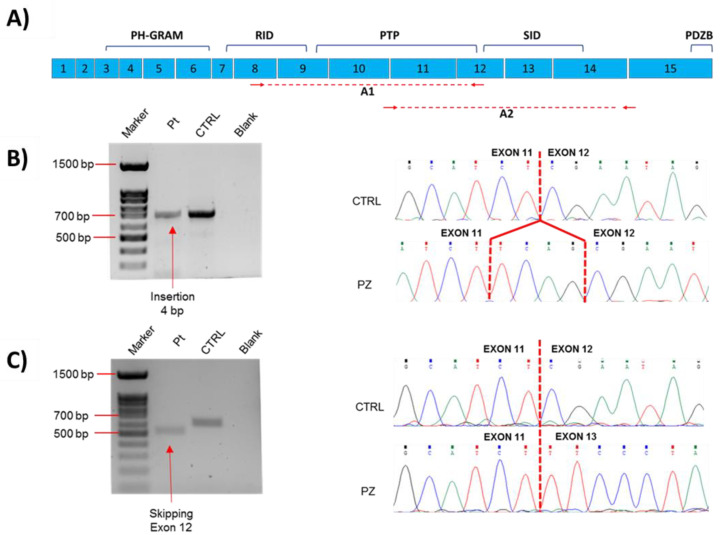
Genetic analyses of aberrant splicing mechanism caused by the novel *MTM1* splicing variant. (**A**) Schematic representation of *MTM1* mRNA showing different protein domains; amplicons (A1 and A2) used for detecting aberrant transcripts are underlined with red dashed line, and primers used for generating these different amplicons are represented with red arrows. (**B**,**C**) Agarose gel electrophoresis and Sanger sequencing of cDNA from muscle showing aberrant transcripts in patient compared to control. (**B**) Amplicon A1 (encompassing exon 8 to 12). It is not possible to appreciate the insertion of 4 bp because amplicon size in the patient (≈702 bp) is similar to the control’s one (≈698 bp). (**C**) Amplicon A2 (encompassing exon 11 to 14). Amplicon size in the patient is lower (≈498 bp) than in the control (≈591 bp) because of the skipping of exon 12. Gel electrophoresis run of Amplicon A2 did not show both aberrant transcripts simultaneously, probably because the transcript with exon 12 skipping is much more abundant than the transcript with inclusion of 4 nucleotides, which, for this reason, is detected only using specific primers. **Lane 1:** 100 bp DNA ladder marker; **Lane 2:** Patient; **Lane 3:** Control; **Lane 4:** Blank.

**Table 1 ijms-23-10274-t001:** Primers used for generating different *MTM1* cDNA amplicons.

AMPLICON	PRIMER SEQUENCE (5′-3′)
Amplicon A1	Fw: GTTCCGTATCGTGCCTCAG
	Rev: GGAGAACGGTCAGCATCGG
Amplicon A2	Fw: CATATCAAGCTCGTTTTGACAG
	Rev: GGATTCGGCTGTTGTTGCTTG

## Data Availability

Data supporting the findings of this study not presented within the article are available upon request.

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
