# Peer review of "Novel Splicing Mutation in MTM1 Leading to Two Abnormal Transcripts Causes Severe Myotubular Myopathy"

_ijms, 2022, doi:10.3390/ijms231810274_

Round 1

Reviewer 1 Report

X-linked myotubular myopathy (XLMTM) is a clinically well-defined severe form of centronuclear myopathy, characterized by generalized weakness and respiratory insufficiency, caused genetic variants/mutations in MTM1 gene.

Authors in this manuscript identified the novel MTM1 c.1261-5T>G intronic variant which interferes with the normal splicing process generating two different abnormal transcripts simultaneously expressed in a three months old male child with hemizygous MTM1 genotype and XLMTM phenotype. The first aberrant transcript, induced by the activation of a cryptic splice site in intron 11, includes four intronic nucleotides upstream of exon 12 resulting in a shift in the transcript reading frame and introducing a new premature stop codon in the catalytic domain of the protein (p.Arg421Serfs*7). The second aberrant MTM1 transcript, with the complete skipping of exon 12. My comments are below:

1). This report expanded the genotypic spectrum of XLMTM underlying the importance of intron-exons boundaries in male patients affected by XLMTM, using a NGS approach.  The manuscript is well organized and written, clinical presentation and diagnostic procedures for this patient are clearly described.

2). Several comments for this manuscript. All figures and Tables should be moved to the end of manuscript to avoid content interruptions caused by line numbers in the main-text.

3). Regarding the molecular data, for the first aberrant transcript, the resultant transcript with premature stop codon (PTC) in the catalytic domain of the MTM1 protein (p.Arg421Serfs*7), which is genetically equivalent to the most common causative MTM1 genetic mutation of XLMTM (p.Arg421Ter). It has been well defined that this nonsense mutation of MTM1 results in total loss of myotubularin expression due to NMD (nonsense-mediated decay). However, this aberrant transcript was found to be quite abundant in transcriptional level (Northern blot or just RT-PCR products on gels), compared to WT (Fig. 3B). In order to be consistent to previous observations, a Western blot has to be performed to demonstrate the diminished MTM1 at protein level (there are many validated MTM1 antibodies).

4). Based on the genomic positions of primer sets provided by authors, gel electrophoresis of Amplicon A1 (702 nts) can only be observed in Fig. 3B, however, gel electrophoresis of Amplicon A1 (595 nts) and A2 (498 nts) should be observed simultaneously in Fig. 3C, please explain why there is only one band (A2)?

5). If the notion that the majority of intronic mutations in MTM1 gene are frequently related to a more severe form of XLMTM is correct, then only the second aberrant transcript (with a in frame deletion of the region between Arg421 and Gln451) would contribute to this severe form of XLMTM (the first one will be degraded) if author really try to make genotype/phenotype correlation, please add a few sentence in discussion section for this issue.

6). To quantitatively measure the expression levels of two aberrant transcripts compared to normal WT transcript, I would suggest authors to utilize TagMan RT-qPCR approach to simultaneously quantify the relative expression levels of all three possible transcripts of MTM1 gene.

Reviewer 2 Report

The introduction could be more detailed especially the description of molecular processes with MTM1 invovement. Decribing the severety of MTM1 myopathy (4) is not the best reference. Reference (6) would not be the best for explaining the effect of MTM1 on muscle cell differentiation.

Part "clinical findings" should be definitely revised by a native English speaker.

The histological pictures (figure 1) may be improved by an enlargement to see the features decribed in the legend of figure 1.

Figure 2A/B: There are line numbers on the left side of the drawing. Please mark the new cryptic splice site in the electropherogram of the shown sequence.

Round 2

Reviewer 1 Report

In the revised version, authors did some efforts to mostly address my comments/suggestions.